# An Imitation from Observation Approach to Transfer Learning with Dynamics Mismatch

**Siddarth Desai**[§]
Department of Mechanical Engineering
The University of Texas at Austin
`sidrdesai@utexas.edu`

**Ishan Durugkar** [§]
Department of Computer Science
The University of Texas at Austin
`ishand@cs.utexas.edu`

**Haresh Karnan** [§]
Department of Mechanical Engineering
The University of Texas at Austin
`haresh.miriyala@utexas.edu`

**Garrett Warnell**
Army Research Laboratory
`garrett.a.warnell.civ@mail.mil`

**Josiah P. Hanna** [*]
School of Informatics
The University of Edinburgh
`josiah.hanna@ed.ac.uk`

**Peter Stone**
Department of Computer Science
The University of Texas at Austin
and Sony AI
`pstone@cs.utexas.edu`

## Abstract

We examine the problem of transferring a policy learned in a source environment to a target environment with different dynamics, particularly in the case where it is critical to reduce the amount of interaction with the target environment during learning. This problem is particularly important in sim-to-real transfer because simulators inevitably model real-world dynamics imperfectly. In this paper, we show that one existing solution to this transfer problem—*grounded action transformation*—is closely related to the problem of *imitation from observation* (IfO): learning behaviors that mimic the observations of behavior demonstrations. After establishing this relationship, we hypothesize that recent state-of-the-art approaches from the IfO literature can be effectively repurposed for grounded transfer learning. To validate our hypothesis we derive a new algorithm—generative adversarial reinforced action transformation (GARAT)—based on adversarial imitation from observation techniques. We run experiments in several domains with mismatched dynamics, and find that agents trained with GARAT achieve higher returns in the target environment compared to existing black-box transfer methods.

## 1 Introduction

Transfer learning with dynamics mismatch refers to using experience in a source environment to more efficiently learn control policies that perform well in a target environment, where the two environments differ only in their transition dynamics. For example, if the friction coefficient in the source and target environments is sufficiently different it might cause the action of placing a foot on the ground to work well in one environment, but cause the foot to slip in the other. One possible application of such transfer is where the source environment is a simulator and the target environment

---

[*]to be joining the Computer Sciences department at the University of Wisconsin – Madison
[§]Equal contribution

is a robot in the real world, called sim-to-real. In sim-to-real scenarios, source environment (simulator) experience is readily available, but target environment (real world) experience is expensive. Sim-to-real transfer has been used effectively to learn a fast humanoid walk [15], dexterous manipulation [29, 22, 38, 26, 6, 24, 23], and agile locomotion skills [32]. In this work, we focus on the paradigm of simulator grounding [10, 15, 8], which modifies the source environment's dynamics to more closely match the target environment dynamics using a relatively small amount of target environment data. Policies then learned in such a grounded source environment transfer better to the target environment.

Separately, the machine learning community has also devoted attention to imitation learning [5], i.e. the problem of learning a policy to mimic demonstrations provided by another agent. In particular, recent work has considered the specific problem of *imitation from observation* (IfO) [25], in which an imitator mimics the expert's behavior without knowing which actions the expert took, only the outcomes of those actions (i.e. state-only demonstrations). While the lack of action information presents an additional challenge, recently-proposed approaches have suggested that this challenge may be addressable [48, 50].

In this paper, we show that a particular grounded transfer technique that has been shown to successfully accomplish sim-to-real transfer, called *grounded action transformation* (GAT) [15], can be seen as a form of IfO. We therefore hypothesize that recent, state-of-the-art approaches for addressing the IfO problem might also be effective for grounding the source environment, leading to improved transfer. Specifically, we derive a distribution-matching objective similar to ones used in adversarial approaches for generative modeling [14], imitation learning [18], and IfO [49] with considerable empirical success. Based on this objective, we propose a novel algorithm, Generative Adversarial Reinforced Action Transformation (GARAT), to ground the source environment by reducing the distribution mismatch between the source and target environments.

Our experiments confirm our hypothesis by showing that GARAT reduces the difference in the dynamics between two environments more effectively than GAT. Moreover, our experiments show that, in several domains, this improved grounding translates to better transfer of policies from one environment to the other.

The contributions of this paper are as follows: *(1)* we show that learning the grounded action transformation can be seen as an *IfO* problem, *(2)* we derive a novel adversarial imitation learning algorithm, GARAT, to learn an action transformation policy for transfer learning with dynamics mismatch, and *(3)* we experimentally evaluate the efficacy of GARAT for transfer with dynamics mismatch.

## 2  Background

We begin by introducing notation, reviewing the transfer learning with dynamics mismatch problem formulation, and describing the action transformation approach for sim-to-real transfer. We also provide a brief overview of imitation learning and imitation from observation.

### 2.1  Notation

We consider here sequential decision processes formulated as Markov decision processes (MDPs) [42]. An MDP $\mathcal{M}$ is a tuple $\langle \mathcal{S}, \mathcal{A}, R, P, \gamma, \rho_0 \rangle$ consisting of a set of states, $\mathcal{S}$; a set of actions, $\mathcal{A}$; a reward function, $R : \mathcal{S} \times \mathcal{A} \times \mathcal{S} \longmapsto \Delta([r_{\min}, r_{\max}])$ (where $\Delta([r_{\min}, r_{\max}])$ denotes a distribution over the interval $[r_{\min}, r_{\max}] \subset \mathbb{R}$); a discount factor, $\gamma \in [0, 1)$; a transition function, $P : \mathcal{S} \times \mathcal{A} \longmapsto \Delta(\mathcal{S})$; and an initial state distribution, $\rho_0 : \Delta(\mathcal{S})$. An RL agent uses a policy $\pi : \mathcal{S} \longmapsto \Delta(\mathcal{A})$ to select actions in the environment. In an environment with transition function $P \in \mathcal{T}$, the agent aims to learn a policy $\pi \in \mathbf{\Pi}$ to maximize its expected discounted return $\mathbb{E}_{\pi,P}[G_0] = \mathbb{E}_{\pi,P}\left[\sum_{t=0}^{\infty} \gamma^t R_t\right]$, where $R_t \sim R(s_t, a_t, s_{t+1})$, $s_{t+1} \sim P(s_t, a_t)$, $a_t \sim \pi(s_t)$, and $s_0 \sim \rho_0$.

Given a fixed $\pi$ and a specific transition function $P_q$, the marginal transition distribution is $\rho_q(s, a, s') := (1 - \gamma)\pi(a|s)P_q(s'|s, a)\sum_{t=0}^{\infty} \gamma^t p(s_t = s|\pi, P_q)$ where $p(s_t = s|\pi, P_q)$ is the probability of being in state $s$ at time $t$. The marginal transition distribution is the probability of being in state $s$ marginalized over time $t$, taking action $a$ under policy $\pi$, and ending up in state $s'$ under transition function $P_q$ (laid out more explicitly in Appendix A). We can denote the expected return

under a policy $\pi$ and a transition function $P_q$ in terms of this marginal distribution as:

$$\mathbb{E}_{\pi,q}[G_0] = \frac{1}{(1-\gamma)} \sum_{s,a,s'} \rho_q(s,a,s')R(s'|s,a) \tag{1}$$

## 2.2 Transfer Learning with Dynamics Mismatch and Grounded Action Transformation

Let $P_s, P_t \in \mathcal{T}$ be the transition functions for two otherwise identical MDPs, $\mathcal{M}_s$ and $\mathcal{M}_t$, representing the source and target environments respectively. Transfer learning with dynamics mismatch, as opposed to transfer learning in general, aims to train an agent policy to maximize return in $\mathcal{M}_t$ with limited trajectories from $\mathcal{M}_t$, and as many as needed in $\mathcal{M}_s$.

The work presented here is specifically concerned with a particular class of approaches used in sim-to-real transfer known as simulator grounding approaches [1, 8, 10]. Here the source environment is the simulator and the target environment is the real world. These approaches use some interactions with the target environment to *ground* the source environment dynamics to more closely match the target environment dynamics. Because it may sometimes be difficult or impossible to modify the source environment itself (when the source environment is a black-box simulator, for example), the recently-proposed grounded action transformation (GAT) approach [15] seeks to instead induce grounding by modifying the agent's actions before using them in the source environment. This modification is accomplished via an action transformation function $\pi_g : \mathcal{S} \times \mathcal{A} \longmapsto \Delta(\mathcal{A})$ that takes as input the state and action of the agent, and produces an action to be presented to the source environment. From the agent's perspective, composing the action transformation with the source environment changes the source environment's transition function. We call this modified source environment the *grounded* environment, and its transition function is given by

$$P_g(s'|s,a) = \sum_{\tilde{a}\in\mathcal{A}} P_s(s'|s,\tilde{a})\pi_g(\tilde{a}|s,a) \tag{2}$$

The action transformation approach aims to learn function $\pi_g \in \mathbf{\Pi}_g$ such that the resulting transition function $P_g$ is as close as possible to $P_t$. We denote the marginal transition distributions in the source and target environments by $\rho_s$ and $\rho_t$ respectively, and $\rho_g \in \mathcal{P}_g$ for the grounded environment.

GAT learns a model of the target environment dynamics $\hat{P}_t(s'|s,a)$, an inverse model of the source environment dynamics $\hat{P}_s^{-1}(a|s,s')$, and uses the composition of the two as the action transformation function, i.e. $\pi_g(\tilde{a}|s,a) = \hat{P}_s^{-1}(\tilde{a}|s, \hat{P}_t(s'|s,a))$.

## 2.3 Imitation Learning

In parallel to advances in sim-to-real transfer, the machine learning community has also made considerable progress on the problem of imitation learning. Imitation learning [5, 36, 39] is the problem setting where an agent tries to mimic trajectories $\{\xi_0, \xi_1, \ldots\}$ where each $\xi$ is a demonstrated trajectory $\{(s_0, a_0), (s_1, a_1), \ldots\}$ induced by an expert policy $\pi_{exp}$.

Various methods have been proposed to address the imitation learning problem. Behavioral cloning [4] uses the expert's trajectories as labeled data and uses supervised learning to recover the maximum likelihood policy. Another approach instead relies on reinforcement learning to learn the policy, where the required reward function is recovered using inverse reinforcement learning (IRL) [28]. IRL aims to recover a reward function under which the demonstrated trajectories would be optimal.

A related setting to learning from state-action demonstrations is the imitation from observation (IfO) [25, 30, 48, 49] problem. Here, an agent observes an expert's state-only trajectories $\{\zeta_0, \zeta_1, \ldots\}$ where each $\zeta$ is a sequence of states $\{s_0, s_1, \ldots\}$. The agent must then learn a policy $\pi(a|s)$ to imitate the expert's behavior, without being given labels of which actions to take.

## 3 GAT as Imitation from Observation

We now show that the underlying problem of GAT—i.e., learning an action transformation for sim-to-real transfer—can also been seen as an IfO problem. Adapting the definition by Liu et al. [25], an IfO problem is a sequential decision-making problem where the policy imitates state-only trajectories

$\{\zeta_0, \zeta_1, \ldots\}$ produced by a Markov process, with no information about what actions generated those trajectories. To show that the action transformation learning problem fits this definition, we must show that it *(1)* is a sequential decision-making problem and *(2)* aims to imitate state-only trajectories produced by a Markov process, with no information about what actions generated those trajectories.

Starting with *(1)*, it is sufficient to show that the action transformation function is a policy in an MDP [34]. This action transformation MDP can be seen clearly if we combine the target environment MDP and the fixed agent policy $\pi$. Let the joint state and action space $\mathcal{X} := \mathcal{S} \times \mathcal{A}$ with $x := (s, a) \in \mathcal{X}$ be the state space of this new MDP. The combined transition function is $P_s^x(x'|x, \tilde{a}) = P_s(s'|s, \tilde{a})\pi(a'|s')$, where $x' = (s', a')$, and initial state distribution is $\rho_0^x(x) = \rho_0(s)\pi(a|s)$. For completeness, we consider a reward function $R^x : \mathcal{X} \times \mathcal{A} \times \mathcal{X} \longmapsto \Delta([r_{\min}, r_{\max}])$ and discount factor $\gamma_x \in [0, 1)$, which are not essential for an IfO problem. With these components, the action transformation environment is an MDP $\langle \mathcal{X}, \mathcal{A}, R^x, P_s^x, \gamma_x, \rho_0^x \rangle$. The action transformation function $\pi_g(\tilde{a}|s, a)$, now $\pi_g^x(\tilde{a}|x)$, is then clearly a mapping from states to a distribution over actions, i.e. it is a policy in an MDP. Thus, the action transformation learning problem is a sequential decision-making problem.

We now consider the action transformation objective to show *(2)*. When learning the action transformation policy, we have trajectories $\{\tau_0, \tau_1, \ldots\}$, where each trajectory $\tau = \{(s_0, a_0 \sim \pi(s_0)), (s_1, a_1 \sim \pi(s_1)), \ldots\}$ is obtained by sampling actions from agent policy $\pi$ in the target environment. Re-writing $\tau$ in the above MDP, $\tau = \{x_0, x_1, \ldots\}$. If an expert action transformation policy $\pi_g^* \in \mathbf{\Pi}_g$ is capable of mimicking the dynamics of the target environment, $P_t^x(x'|x) = \sum_{\tilde{a} \in \mathcal{A}} P_s^x(x'|x, \tilde{a})\pi_g^*(\tilde{a}|x)$, then we can consider the above trajectories to be produced by a Markov process with dynamics $P_s^x(x'|x, \tilde{a})$ and policy $\pi_g^*(\tilde{a}|x)$. The action transformation aims to imitate the state-only trajectories $\{\tau_0, \tau_1, \ldots\}$ produced by a Markov process, with no information about what actions generated those trajectories.

The problem of learning the action transformation thus satisfies the conditions we identified above, and so it is an IfO problem.

## 4 Generative Adversarial Reinforced Action Transformation

The insight above naturally leads to the following question: *if learning an action transformation for transfer learning is equivalent to IfO, might recently-proposed IfO approaches lead to better transfer learning approaches?* To investigate the answer, we derive a novel generative adversarial approach inspired by GAIfO[49] that can be used to train the action transformation policy using IfO. A source environment grounded with this action transformation policy can then be used to train an agent policy which can be expected to transfer effectively to a given target environment. We call our approach generative adversarial reinforced action transformation (GARAT), and Algorithm 1 lays out its details.

The rest of this section details our derivation of the objective used in GARAT. First, in Section 4.1, we formulate a procedure for action transformation using a computationally expensive IRL step to extract a reward function and then learning an action transformation policy based on that

---

**Algorithm 1** GARAT

**Input:** Target environment with $P_t$, source environment with $P_s$, number of update steps $N$
  Agent policy $\pi$ with parameters $\eta$, pretrained in source environment;
  Initialize action transformation policy $\pi_g$ with parameters $\theta$
  Initialize discriminator $D_\phi$ with parameters $\phi$
  **while** *performance of policy $\pi$ in target environment not satisfactory* **do**
    Rollout policy $\pi$ in target environment to obtain trajectories $\{\tau_{t,1}, \tau_{t,2}, \ldots\}$
    **for** $i = 0, 1, 2, \ldots N$ **do**
      Rollout Policy $\pi$ in grounded source environment and obtain trajectories $\{\tau_{g,1}, \tau_{g,2}, \ldots\}$
      Update parameters $\phi$ of $D_\phi$ using gradient descent to minimize
      $-\left(\mathbb{E}_{\tau_g}[\log(D_\phi(s, a, s'))] + \mathbb{E}_{\tau_t}[\log(1 - D_\phi(s, a, s'))]\right)$
      Update parameters $\theta$ of $\pi_g$ using policy gradient with reward $-[\log D_\phi(s, a, s')]$
    **end**
    Optimize parameters $\eta$ of $\pi$ in source environment grounded with action transformer $\pi_g$
  **end**

---

reward. Then, in Section 4.2, we show that this entire procedure is equivalent to directly reducing the marginal transition distribution discrepancy between the target environment and the grounded source environment. This is important, as recent work [14, 18, 49] has shown that adversarial approaches are a promising algorithmic paradigm to reduce such discrepancies. Thus, in Section 4.3, we explicitly formulate a generative adversarial objective upon which we build the proposed approach.

## 4.1 Action Transformation Inverse Reinforcement Learning

We first lay out a procedure to learn the action transformation policy by extracting the appropriate cost function, which we term action transformation IRL (ATIRL). We use the cost function formulation in our derivation, similar to previous work [18, 49]. ATIRL aims to identify a cost function such that the observed target environment transitions yield higher return than any other possible transitions. We consider the set of cost functions $\mathcal{C}$ as all functions $\mathbb{R}^{\mathcal{S} \times \mathcal{A} \times \mathcal{S}} = \{c : \mathcal{S} \times \mathcal{A} \times \mathcal{S} \longmapsto \mathbb{R}\}$.

$$\texttt{ATIRL}_\psi(P_t) := \operatorname*{argmax}_{c \in \mathcal{C}} -\psi(c) + \left( \min_{\pi_g \in \mathbf{\Pi}_g} \mathbb{E}_{\rho_g}[c(s, a, s')] \right) - \mathbb{E}_{\rho_t}[c(s, a, s')] \qquad (3)$$

where $\psi : \mathbb{R}^{\mathcal{S} \times \mathcal{A} \times \mathcal{S}} \longmapsto \overline{\mathbb{R}}$ is a (closed, proper) convex reward function regularizer, and $\overline{\mathbb{R}}$ denotes the extended real numbers $\mathbb{R} \bigcup \{\infty\}$. This regularizer is used to avoid overfitting the expressive set $\mathcal{C}$. Note that $\pi_g$ influences $\rho_g$ (Equation 10 in Appendix A) and $P_t$ influences $\rho_t$. Similar to GAIfO, we do not use causal entropy in our ATIRL objective due to the surjective mapping from $\mathbf{\Pi}_g$ to $\mathcal{P}_g$.

The action transformation then uses this per-step cost function as a reward function in an RL procedure: $\texttt{RL}(c) := \operatorname{argmin}_{\pi_g \in \mathbf{\Pi}_g} \mathbb{E}_{\rho_g}[c(s, a, s')]$. We assume here for simplicity that there is an action transformation policy that can mimic the target environment dynamics perfectly. That is, there exists a policy $\pi_g \in \mathbf{\Pi}_g$, such that $P_g(s'|s, a) = P_t(s'|s, a) \forall s \in \mathcal{S}, a \in \mathcal{A}$. We denote the RL procedure applied to the cost function recovered by ATIRL as $\texttt{RL} \circ \texttt{ATIRL}_\psi(P_t)$.

## 4.2 Characterizing the Policy Induced by ATIRL

This section shows that it is possible to bypass the ATIRL step and learn the action transformation policy directly from data. We show that $\psi$-regularized $\texttt{RL} \circ \texttt{ATIRL}_\psi(P_t)$ implicitly searches for policies that have a marginal transition distribution close to the target environment's, as measured by the convex conjugate of $\psi$, which we denote as $\psi^*$. As a practical consequence, we will then be able to devise a method for minimizing this divergence through the use of generative adversarial techniques in Section 4.3. But first, we state our main theoretical claim:

**Theorem 1.** $\texttt{RL} \circ \texttt{ATIRL}_\psi(P_t)$ *and* $\operatorname{argmin}_{\pi_g} \psi^*(\rho_g - \rho_t)$ *induce policies that have the same marginal transition distribution,* $\rho_g$.

To reiterate, the agent policy $\pi$ is fixed. So the only decisions affecting the marginal transition distributions are of the action transformation policy $\pi_g$. We can now state the following proposition:

**Proposition 4.1.** *For a given* $\rho_g$ *generated by a fixed policy* $\pi$, $P_g$ *is the only transition function whose marginal transition distribution is* $\rho_g$.

Proof in Appendix B.1. We can also show that if two transition functions are equal, then the optimal policy in one will be optimal in the other.

**Proposition 4.2.** *If* $P_t = P_g$, *then* $\operatorname{argmax}_{\pi \in \mathbf{\Pi}} \mathbb{E}_{\pi, P_g}[G_0] = \operatorname{argmax}_{\pi \in \mathbf{\Pi}} \mathbb{E}_{\pi, P_t}[G_0]$.

Proof in Appendix B.2. We now prove Theorem 1, which characterizes the policy learned by $\texttt{RL}(\tilde{c})$ on the cost function $\tilde{c}$ recovered by $\texttt{ATIRL}_\psi(P_t)$.

*Proof of Theorem 1.* To prove Theorem 1, we prove that $\texttt{RL} \circ \texttt{ATIRL}_\psi(P_t)$ and $\operatorname{argmin}_{\pi_g} \psi^*(\rho_g - \rho_t)$ result in the same marginal transition distribution. This proof has three parts, two of which are proving that both objectives above can be formulated as optimizing over marginal transition distributions. The third is to show that these equivalent objectives result in the same distribution.

The output of both $\texttt{RL} \circ \texttt{ATIRL}_\psi(P_t)$ and $\operatorname{argmin}_{\pi_g} \psi^*(\rho_g - \rho_t)$ are policies. To compare the marginal distributions, we first establish a different $\overline{\texttt{RL}} \circ \overline{\texttt{ATIRL}}_\psi(P_t)$ objective that we argue has the same

marginal transition distribution as $\mathtt{RL} \circ \mathtt{ATIRL}_\psi(P_t)$. We define

$$\overline{\mathtt{ATIRL}}_\psi(P_t) := \operatorname*{argmax}_{c \in \mathcal{C}} -\psi(c) + \left( \min_{\rho_g \in \mathcal{P}_g} \mathbb{E}_{\rho_g}\left[c(s,a,s')\right] \right) - \mathbb{E}_{\rho_t}\left[c(s,a,s')\right] \qquad (4)$$

with the same $\psi$ and $\mathcal{C}$ as Equation 3, and similar except the internal optimization for Equation 3 is over $\pi_g \in \mathbf{\Pi}_g$, while it is over $\rho_g \in \mathcal{P}_g$ for Equation 4. We define an RL procedure $\overline{\mathtt{RL}}(\overline{c}) := \operatorname{argmin}_{\rho_g \in \mathcal{P}_g} \mathbb{E}_{\rho_g} c(s,a,s')$ that returns a marginal transition distribution $\rho_g \in \mathcal{P}_g$ which minimizes the given cost function $\overline{c}$. $\overline{\mathtt{RL}}(\overline{c})$ will output the marginal transition distribution $\overline{\rho}_g$.

**Lemma 4.1.** $\overline{\mathtt{RL}} \circ \overline{\mathtt{ATIRL}}_\psi(P_t)$ *outputs a marginal transition distribution* $\overline{\rho}_g$ *which is equal to* $\tilde{\rho}_g$ *induced by* $\mathtt{RL} \circ \mathtt{ATIRL}_\psi(P_t)$.

Proof in Appendix B.3. The mapping from $\mathbf{\Pi}_g$ to $\mathcal{P}_g$ is not injective, and there could be multiple policies $\pi_g$ that lead to the same marginal transition distribution. The above lemma is sufficient for proof of Theorem 1, however, since we focus on the effect of the policy on the transitions.

**Lemma 4.2.** $\overline{\mathtt{RL}} \circ \overline{\mathtt{ATIRL}}_\psi(P_t) = \operatorname{argmin}_{\rho_g \in \mathcal{P}_g} \psi^*(\rho_g - \rho_t)$.

The proof in Appendix B.4 relies on the optimal cost function and the optimal policy forming a saddle point, $\psi^*$ leading to a minimax objective, and these objectives being the same.

**Lemma 4.3.** *The marginal transition distribution of* $\operatorname{argmin}_{\pi_g} \psi^*(\rho_g - \rho_t)$ *is equal to* $\operatorname{argmin}_{\rho_g \in \mathcal{P}_g} \psi^*(\rho_g - \rho_t)$.

Proof in appendix B.5. With these three lemmas, we have proved that $\mathtt{RL} \circ \mathtt{ATIRL}_\psi(P_t)$ and $\operatorname{argmin}_{\pi_g} \psi^*(\rho_g - \rho_t)$ induce policies that have the same marginal transition distribution. $\qquad \square$

Theorem 1 thus tells us that the objective $\operatorname{argmin}_{\pi_g} \psi^*(\rho_g - \rho_t)$ is equivalent to the procedure from Section 4.1. In the next section, we choose a function $\psi$ which leads to our adversarial objective.

### 4.3 Forming the Adversarial Objective

Section 4.2 laid out the objective we want to minimize. To solve $\operatorname{argmin}_{\pi_g} \psi^*(\rho_g - \rho_t)$ we require an appropriate regularizer $\psi$. GAIL [18] and GAIfO [49] optimize similar objectives and have shown a regularizer similar to the following to work well:

$$\psi(c) = \begin{cases} \mathbb{E}_t[g(c(s,a,s'))] & \text{if } c < 0 \\ +\infty & \text{otherwise} \end{cases} \text{ where } g(x) = \begin{cases} -x - log(1-e^x) & \text{if } x < 0 \\ +\infty & \text{otherwise} \end{cases} \qquad (5)$$

It is closed, proper, convex and has a convex conjugate leading to the following minimax objective:

$$\min_{\pi_g \in \mathbf{\Pi}_g} \psi^*(\rho_g - \rho_t) = \min_{\pi_g \in \mathbf{\Pi}_g} \max_D \mathbb{E}_{P_g}[\log(D(s,a,s'))] + \mathbb{E}_{P_t}[\log(1 - D(s,a,s'))] \qquad (6)$$

where the reward for the action transformer policy $\pi_g$ is $-[\log(D(s,a,s'))]$, and $D : \mathcal{S} \times \mathcal{A} \times \mathcal{S} \longmapsto (0,1)$ is a discriminative classifier. These properties have been shown in previous works [18, 49]. Algorithm 1 lays out the steps for learning the action transformer using the above procedure, which we call generative adversarial reinforced action transformation (GARAT).

## 5 Related Work

While our work lies in the space of transfer learning with dynamics mismatch, the eventual goal of this research is to enable effective sim-to-real transfer. In this section, we discuss the variety of sim-to-real methods, work more closely related to GARAT, and some related methods in the IfO literature. Sim-to-real transfer can be improved by making the agent's policy more robust to variations in the environment or by making the simulator more accurate w.r.t. the real world. The first approach, which we call policy robustness methods, encompasses algorithms that train a robust policy that performs well on a range of environments [20, 31, 32, 33, 35, 37, 45, 46]. Robust adversarial reinforcement learning (RARL) [33] is such an algorithm that learns a policy robust to adversarial perturbations

[43]. While primarily focused on training with a modifiable simulator, a version of RARL treats the simulator as a black-box by adding the adversarial perturbation directly to the protagonist's action. Additive noise envelope (ANE) [21] is another black-box robustness method which adds an envelope of Gaussian noise to the agent's action during training.

The second approach, known as domain adaption or system identification, grounds the simulator using real world data to make its transitions more realistic. Since hand engineering accurate simulators [44, 52] can be expensive and time consuming, real world data can be used to adapt low-fidelity simulators to the task at hand. Most simulator adaptation methods [1, 8, 10, 19] rely on access to a parameterized simulator.

GARAT, on the other hand, does not require a modifiable simulator and relies on an action transformation policy applied in the source environment to bring its transitions closer to the target environment. GAT[15] learns an action transformation function similar to GARAT. It was shown to have successfully learned and transferred one of the fastest known walk policies on the humanoid robot, Nao.

GARAT draws from recent generative adversarial approaches to imitation learning (GAIL [18]) and IfO (GAIfO [49]). AIRL[11], FAIRL[13], and WAIL[51] are related approaches which use different divergence metrics to reduce the marginal distribution mismatch. GARAT can be adapted to use any of these metrics, as we show in the appendix.

One of the insights of this paper is that grounding the simulator using action transformation can be seen as a form of IfO. BCO [48] is an IfO technique that utilizes behavioral cloning. I2L [12] is an *IfO* algorithm that aims to learn in the presence of transition dynamics mismatch in the expert and agent's domains, but requires millions of real world interactions to be competent.

# 6 Experiments

In this section, we conduct experiments to verify our hypothesis that GARAT leads to improved transfer in the presence of dynamics mismatch compared to previous methods. We also show that it leads to better source environment grounding compared to the previous action transformation approach, GAT.

We validate GARAT for transfer by transferring the agent policy between Open AI Gym [7] simulated environments with different transition dynamics. We highlight the Minitaur domain (Figure 2) as a particularly useful test since there exist two simulators, one of which has been carefully engineered for high fidelity to the real robot [44]. For other environments, the target environment is the source environment modified in different ways such that a policy trained in the source environment does not transfer well to the target environment. Details of these modifications are provided in Appendix C.1. Apart from a thorough evaluation across multiple different domains, this setup also allows us to compare GARAT and other algorithms against a policy trained directly in the target environment with millions of interactions, which is otherwise prohibitively expensive on a real robot. This setup also allows us to perform a thorough evaluation of sim-to-real algorithms across multiple different domains. We focus here on answering the following questions :

1. How well does GARAT ground the source environment with respect to the target environment?
2. Does GARAT lead to improved transfer with in the presence of dynamics mismatch, compared to other related methods?

## 6.1 Source Environment Grounding

In Figure 1, we evaluate how well GARAT grounds the source environment to the target environment both quantitatively and qualitatively. This evaluation is in the *InvertedPendulum* domain, where the target environment has a heavier pendulum than the source; implementation details are in Appendix C.1. In Figure 1a, we plot the average error in transitions in source environments grounded with GARAT and GAT with different amounts of target environment data, collected by deploying $\pi$ in the target environment. In Figure 1b we deploy the same policy $\pi$ from the same start state in the different environments (source, target, and grounded source). From both these figures it is evident that GARAT leads to a grounded source environment with lower error on average, and responses

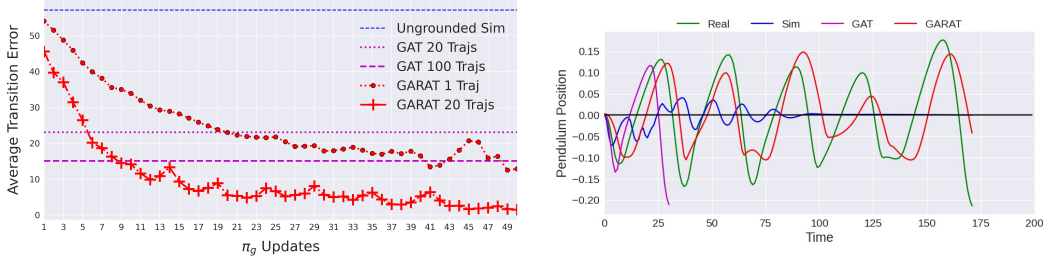

(a) L2 norm of per step transition errors (lower is better) between different source environments and the target environment, shown over number of action transformation policy updates for GARAT.

(b) Example trajectories of the same agent policy deployed in different environments, plotted using the pendulum angle across time. Response of GARAT grounded source environment is the most like target environment.

Figure 1: Evaluation of source environment grounding with GARAT in *InvertedPendulum* domain

qualitatively closer to the target environment compared to GAT. Details of how we obtained these plots are in Appendix C.2.

## 6.2 Transfer Experiments

We now validate the effectiveness of GARAT at transferring a policy from source environment to target environment. For various MuJoCo [47] environments, we pretrain the agent policy $\pi$ in the ungrounded source environment, collect target environment data with $\pi$, use GARAT to ground the source environment, re-train the agent policy until convergence in these grounded source environments, and then evaluate mean return across 50 episodes for the updated agent policy in the target environment.

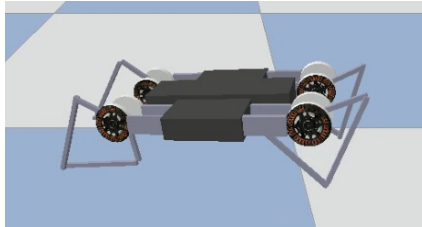

Figure 2: The Minitaur Domain

The agent policy $\pi$ and action transformation policy $\pi_g$ are trained with TRPO [40] and PPO [41] respectively. The specific hyperparameters used are provided in Appendix C. We use the implementations of TRPO and PPO provided in the stable-baselines library [17]. For every $\pi_g$ update, we update the GARAT discriminator $D_\phi$ once as well. Results here use the losses detailed in Algorithm 1. However, we find that GARAT is just as effective with other divergence measures [11, 13, 51] (Appendix C).

GARAT is compared to GAT [15], RARL [33] adapted for a black-box simulator, and action-noise-envelope (ANE) [21]. $\pi_t$ and $\pi_s$ denote policies trained in the target environment and source environment respectively until convergence. We use the best performing hyperparameters for these methods, specified in Appendix C.

Figure 3 shows that, in most of the domains, GARAT with just a few thousand transitions from the target environment facilitates transfer of policies that perform on par with policies trained directly in the target environment using 1 million transitions. GARAT also consistently performs better than previous methods on all domains, except *HopperHighFriction*, where most of the methods perform well. The shaded envelope denotes the standard error across 5 experiments with different random seeds for all the methods. Apart from the MuJoCo simulator, we also show successful transfer in the PyBullet simulator [9] using the *Ant* domain. Here the target environment has gravity twice that of the source environment, resulting in purely source environment-trained policies collapsing ineffectually in the target environment. In this relatively high dimensional domain, as well as in *Walker*, we see GARAT still transfers a competent policy while the related methods fail.

In the Minitaur domain [44] we use the high fidelity simulator as our target environment. Here as well, a policy trained in the source environment does not directly transfer well to the target environment [53]. We see in this realistic setting that GARAT learns a policy that obtains more than 80% of the optimal target environment performance with just 1000 target environment transitions while the next best baseline (GAT) obtains at most 50%, requiring ten times more target environment data.

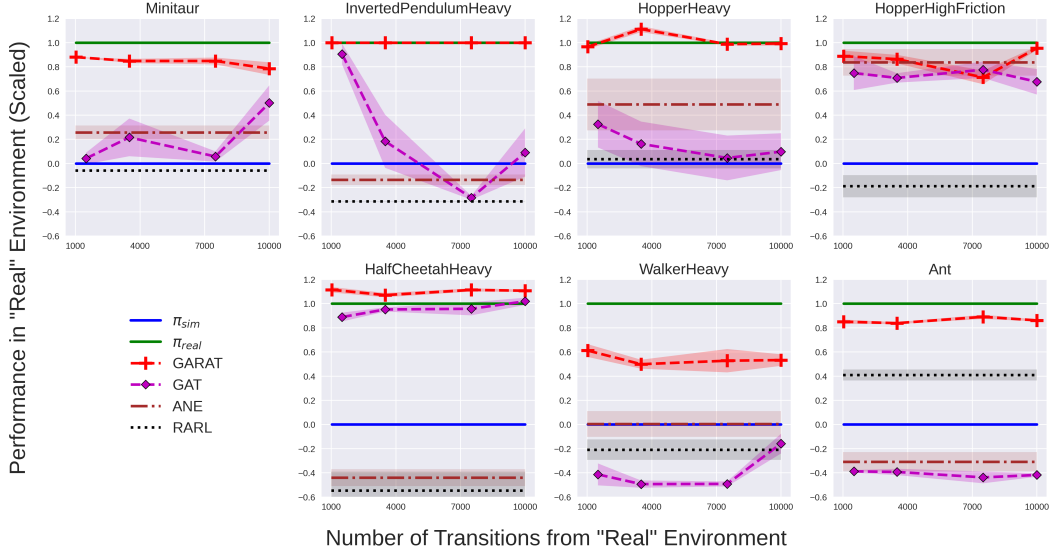

Figure 3: Performance of different techniques evaluated in target environment. Environment return on the $y$-axis is scaled such that $\pi_t$ achieves 1 and $\pi_s$ achieves 0.

# 7 Conclusion

In this paper, we have shown that grounded action transformation, a particular kind of grounded transfer technique, can be seen as a form of imitation from observation. We use this insight to develop GARAT, an adversarial imitation from observation algorithm for grounded transfer. We hypothesized that such an algorithm would lead to improved grounding of the source environment as well as better transfer compared to related techniques. This hypothesis is validated in Section 6 where we show that GARAT leads to better grounding of the source environment as compared to GAT, and improved transfer to the target environment on various mismatched environment transfers, including the realistic Minitaur domain.

## Acknowledgements and Disclosure of Funding

This work has taken place in the Learning Agents Research Group (LARG) at the Artificial Intelligence Laboratory, The University of Texas at Austin. LARG research is supported in part by grants from the National Science Foundation (CPS-1739964, IIS-1724157, NRI-1925082), the Office of Naval Research (N00014-18-2243), Future of Life Institute (RFP2-000), Army Research Office (W911NF-19-2-0333), DARPA, Lockheed Martin, General Motors, and Bosch. The views and conclusions contained in this document are those of the authors alone. Peter Stone serves as the Executive Director of Sony AI America and receives financial compensation for this work. The terms of this arrangement have been reviewed and approved by the University of Texas at Austin in accordance with its policy on objectivity in research.

## Broader Impact

Reinforcement learning [42] is being considered as an effective tool to train autonomous agents in various important domains like robotics, medicine, etc. A major hurdle to deploying learning agents in these environments is the massive exploration and data requirements [16] to ensure that these agents learn effective policies. Real world interactions and exploration in these situations could be extremely expensive (wear and tear on expensive robots), or dangerous (treating a patient in the medical domain).

Sim-to-real transfer aims to address this hurdle and enables agents to be trained mostly in simulation and then transferred to the real world based on very few interactions. Reducing the requirement for

real world data for autonomous agents might open up the viability for autonomous agents in other fields as well.

Improved sim-to-real transfer will also reduce the pressure for high fidelity simulators, which require significant engineering effort [8, 44]. Simulators are also developed with a task in mind, and are generally not reliable outside their specifications. Sim-to-real transfer might enable simulators that learn to adapt to the task that needs to be performed, a potential direction for future research.

Sim-to-real research needs to be handled carefully, however. Grounded simulators might lead to a false sense of confidence in a policy trained in such a simulator. However, a simulator grounded with real world data will still perform poorly in situations outside the data distribution. As has been noted in the broader field of machine learning [3], out of training distribution situations might lead to unexpected consequences. Simulator grounding must be done carefully in order to guarantee that the grounding is applied over all relevant parts of the environment.

Improved sim-to-real transfer could increase reliance on compute and reduce incentives for sample efficient methods. The field should be careful in not abandoning this thread of research as the increasing cost and impact of computation used by machine learning becomes more apparent [2].

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
