[Supplementary Material]

## A  Marginal Distributions and Returns

We expand the marginal transition distribution ($\rho_{sim}$) definition to be more explicit below.

$$\rho_{sim,t}(s, a, s') := \rho_{sim,t}(s)\pi(a|s)P_{sim}(s'|s, a) \tag{7}$$

$$\rho_{sim,t}(s') := \sum_{s \in \mathcal{S}}\sum_{a \in \mathcal{A}} \rho_{sim,t-1}(s, a, s') \tag{8}$$

$$\rho_{sim}(s, a, s') := (1 - \gamma)\sum_{t=0}^{\infty} \gamma^t \rho_{sim,t}(s, a, s') \tag{9}$$

where $\rho_{sim,0}(s) = \rho_0(s)$ is the starting state distribution. Written in a single equation:

$$\rho_{sim}(s, a, s') = (1 - \gamma)\sum_{s_0 \in \mathcal{S}} \rho_0(s_0)\sum_{t=0}^{\infty}\gamma^t \sum_{a_t \in \mathcal{A}}\sum_{s_{t+1} \in \mathcal{S}} \pi(a_t|s_t)P(s_{t+1}|s_t, a_t)$$

The expected return can be written more explicitly to show the dependence on the transition function. It then makes the connection to 1 more explicit.

$$\mathbb{E}_{\pi,P}[G_0] = \mathbb{E}_{\pi,P}\left[\sum_{t=0}^{\infty}\gamma^t R(s_t, a_t, s_{t+1})\right]$$

$$= \sum_{s_0 \in \mathcal{S}} \rho_0(s_0)\sum_{t=0}^{\infty}\gamma^t \sum_{a_t \in \mathcal{A}}\sum_{s_{t+1} \in \mathcal{S}} \pi(a_t|s_t)P(s_{t+1}|s_t, a_t)R(s_t, a_t, s_{t+1})$$

In the grounded simulator, the action transformer policy $\pi_g$ transforms the transition function as specified in Section 2.2. Ideally, such a $\pi_g \in \mathbf{\Pi}_g$ exists. We denote the marginal transition distributions in sim and real by $\rho_{sim}$ and $\rho_{real}$ respectively, and $\rho_g \in \mathcal{P}_g$ for the grounded simulator. The distribution $\rho_g$ relies on $\pi_g \in \mathbf{\Pi}_g$ as follows:

$$\rho_g(s, a, s') = (1 - \gamma)\pi(a|s)\sum_{\tilde{a} \in \mathcal{A}} P_{sim}(s'|s, \tilde{a})\pi_g(\tilde{a}|s, a)\sum_{t=0}^{\infty}\gamma^t p(s_t = s|\pi, P_g) \tag{10}$$

The marginal transition distribution of the simulator after action transformation, $\rho_g(s, a, s')$, differs in Equation 7 as follows:

$$\rho_{g,t}(s, a, s') := \rho_{g,t}(s)\pi(a|s)\sum_{\tilde{a} \in \mathcal{A}} \pi_g(\tilde{a}|s, a)P_g(s'|s, \tilde{a}) \tag{11}$$

## B  Proofs

### B.1  Proof of Proposition 4.1

**Proposition 4.1.** *For a given $\rho_g$ generated by a fixed policy $\pi$, $P_g$ is the only transition function whose marginal transition distribution is $\rho_g$.*

*Proof.* We prove the above statement by contradiction. Consider two transition functions $P_1$ and $P_2$ that have the same marginal distribution $\rho_\pi$ under the same policy $\pi$, but differ in their likelihood for at least one transition $(s, a, s')$.

$$P_1(s'|s, a) \neq P_2(s'|s, a) \tag{12}$$

Let us denote the marginal distributions for $P_1$ and $P_2$ under policy $\pi$ as $\rho_1^\pi$ and $\rho_2^\pi$. Thus, $\rho_1^\pi(s) = \rho_2^\pi(s) \ \forall s \in \mathcal{S}$ and $\rho_1^\pi(s, a, s') = \rho_2^\pi(s, a, s')\forall s, s' \in \mathcal{S}, a \in \mathcal{A}$.

The marginal likelihood of the above transition for both $P_1$ and $P_2$ is:

$$\rho_1^\pi(s,a,s') = \sum_{t=0}^{T-1} \rho_1^\pi(s)\pi(a|s)P_1(s'|s,a)$$

$$\rho_2^\pi(s,a,s') = \sum_{t=0}^{T-1} \rho_2^\pi(s)\pi(a|s)P_2(s'|s,a)$$

Since the marginal distributions match, and the policy is the same, this leads to the equality:

$$P_1(s'|s,a) = P_2(s'|s,a) \forall s, s' \in \mathcal{S}, a \in \mathcal{A} \tag{13}$$

Equation 13 contradicts Equation 12, proving our claim. $\qquad\square$

## B.2 Proof of Proposition 4.2

**Proposition 4.2.** *If $P_{real} = P_g$, then $\mathrm{argmax}_{\pi \in \mathbf{\Pi}} \mathbb{E}_{\pi, P_g}[G_0] = \mathrm{argmax}_{\pi \in \mathbf{\Pi}} \mathbb{E}_{\pi, P_{real}}[G_0]$.*

*Proof.* We overload the notation slightly and refer to $\rho_{real}^\pi$ as the marginal transition distribution in the real world while following agent policy $\pi$. Proposition 4.1 still holds under this expanded notation.

From Proposition 4.1, if $P_{real} = P_g$, we can say that $\rho_{real}^\pi = \rho_g^\pi \forall \pi \in \mathbf{\Pi}$. From Equation 1, $\mathbb{E}_{\pi,g}[G_0] = \mathbb{E}_{\pi,real}[G_0] \forall \pi \in \mathbf{\Pi}$, and $\mathrm{argmax}_{\pi \in \mathbf{\Pi}} \mathbb{E}_{\pi,g}[G_0] = \mathrm{argmax}_{\pi \in \mathbf{\Pi}} \mathbb{E}_{\pi,real}[G_0]$. $\qquad\square$

## B.3 Proof of Lemma 4.1

**Lemma 4.1.** $\overline{\mathrm{RL}} \circ \overline{\mathrm{ATIRL}}_\psi(P_{real})$ *outputs a marginal transition distribution $\overline{\rho}_g$ which is equal to $\tilde{\rho}_g$ induced by* $\mathrm{RL} \circ \mathrm{ATIRL}_\psi(P_{real})$.

*Proof.* For every $\rho_g \in \mathcal{P}_g$, there exists at least one action transformer policy $\pi_g \in \mathbf{\Pi}_g$, from our definition of $\mathcal{P}_g$. Let $\mathrm{RL} \circ \mathrm{ATIRL}_\psi(P_{real})$ lead to a policy $\tilde{\pi}_g$, with a marginal transition distribution $\tilde{\rho}_g$. The marginal transition distribution induced by $\overline{\mathrm{RL}} \circ \overline{\mathrm{ATIRL}}_\psi(P_{real})$ is $\overline{\rho}_g$.

We need to prove that $\tilde{\rho}_g = \overline{\rho}_g$, and we do so by contradiction. We assume that $\tilde{\rho}_g \neq \overline{\rho}_g$. For this inequality to be true, the marginal transition distribution of the result of $\mathrm{RL}(\tilde{c})$ must be different than the result of $\overline{\mathrm{RL}}(\overline{c})$, or the cost functions $\tilde{c}$ and $\overline{c}$ must be different.

Let us compare the $\mathrm{RL}$ procedures first. Assume that $\tilde{c} = \overline{c}$.

$$
\begin{aligned}
\mathrm{RL}(\tilde{c}) &= \underset{\pi}{\mathrm{argmin}}\, \mathbb{E}_{\rho_g}\left[\tilde{c}(s,a,s')\right] \\
&= \underset{\rho_g}{\mathrm{argmin}}\, \mathbb{E}_{\rho_g}\left[\tilde{c}(s,a,s')\right] \quad \texttt{...(surjective mapping)} \\
&= \overline{\mathrm{RL}}(\overline{c}) (\tilde{\mathtt{c}} = \overline{\mathtt{c}})
\end{aligned}
$$

which leads to a contradiction.

Now let's consider the cost functions presented by $\mathrm{ATIRL}_\psi(P_{real})$ and $\overline{\mathrm{ATIRL}}_\psi(P_{real})$. Since $\mathrm{RL}(\tilde{c})$ and $\overline{\mathrm{RL}}(\overline{c})$ lead to the same marginal transition distributions, for the inequality we assumed at the beginning of this proof to be true, $\mathrm{ATIRL}_\psi(P_{real})$ and $\overline{\mathrm{ATIRL}}_\psi(P_{real})$ must return different cost functions.

$$\texttt{ATIRL}_\psi(P_{real}) = \operatorname*{argmax}_{c \in \mathcal{C}} -\psi(c) + \left( \min_{\pi_g} \mathbb{E}_{P_g}[c(s,a,s')] \right) - \mathbb{E}_{P_{real}}[c(s,a,s')]$$

$$= \operatorname*{argmax}_{c \in \mathcal{C}} -\psi(c) + \left( \min_{\pi_g} \sum_{s,a,s'} \rho_g(s,a,s')c(s,a,s') \right) -$$

$$\sum_{s,a,s'} \rho_{real}(s,a,s')c(s,a,s')$$

$$= \operatorname*{argmax}_{c \in \mathcal{C}} -\psi(c) + \left( \min_{\rho_g} \sum_{s,a,s'} \rho_g(s,a,s')c(s,a,s') \right) -$$

$$\sum_{s,a,s'} \rho_{real}(s,a,s')c(s,a,s')$$

$$= \overline{\texttt{ATIRL}}_\psi(P_{real})$$

which leads to another contradiction. Therefore, we can say that $\overline{\rho}_g = \rho_{\tilde{g}}$. $\qquad\square$

## B.4 Proof of Lemma 4.2

We prove convexity under a particular agent policy $\pi$ but across AT policies $\pi_g \in \mathbf{\Pi}_g$

**Lemma B.1.** $\mathcal{P}_g$ *is compact and convex.*

*Proof.* We first prove convexity of $\rho_{\mathbf{\Pi}_g,t}$ for $\pi_g \in \mathbf{\Pi}_g$ and $0 \le t < \infty$, by means of induction.

Base case: $\lambda\rho_{at_1,0} + (1-\lambda)\rho_{at_2,0} \in \rho_{\mathbf{\Pi}_g,0}$, for $0 \le \lambda \le 1$.

$$\lambda\rho_{at_1,0}(s,a,s') + (1-\lambda)\rho_{at_2,0}(s,a,s') = \lambda\rho_0(s)\pi(a|s)\sum_{\tilde{a}\in\mathcal{A}}\pi_{at_1}(\tilde{a}|s,a)P_{sim}(s'|s,\tilde{a})$$

$$+ (1-\lambda)\rho_0(s)\pi(a|s)\sum_{\tilde{a}\in\mathcal{A}}\pi_{at_2}(\tilde{a}|s,a)P_{sim}(s'|s,\tilde{a})$$

$$= \rho_0(s)\pi(a|s)\sum_{\tilde{a}\in\mathcal{A}}\left(\lambda\pi_{at_1}(\tilde{a}|s,a) + (1-\lambda\pi_{at_2}(\tilde{a}|s,a))\right)P_{sim}(s'|s,\tilde{a})$$

$\mathbf{\Pi}_g$ is convex and hence $\rho_0(s)\pi(a|s)\sum_{\tilde{a}\in\mathcal{A}}\left(\lambda\pi_{at_1}(\tilde{a}|s,a) + (1-\lambda\pi_{at_2}(\tilde{a}|s,a))\right)P_{sim}(s'|s,\tilde{a})$ is a valid distribution, meaning $\rho_{\mathbf{\Pi}_g,0}$ is convex.

Induction Step: If $\rho_{\mathbf{\Pi}_g,t-1}$ is convex, $\rho_{\mathbf{\Pi}_g,t}$ is convex.

If $\rho_{\mathbf{\Pi}_g,t-1}$ is convex, $\lambda\rho_{at_1,t}(s) + (1-\lambda)\rho_{at_2,t}(s)$ is a valid distribution. This is true simply by summing the distribution at time $t-1$ over states and actions.

$$\lambda\rho_{at_1,t}(s,a,s') + (1-\lambda)\rho_{at_2,t}(s,a,s') = \lambda\rho_{at_1,t}(s)\pi(a|s)\sum_{\tilde{a}\in\mathcal{A}}\pi_{at_1}(\tilde{a}|s,a)P_{sim}(s'|s,\tilde{a})$$

$$+ (1-\lambda)\rho_{at_2,t}(s)\pi(a|s)\sum_{\tilde{a}\in\mathcal{A}}\pi_{at_2}(\tilde{a}|s,a)P_{sim}(s'|s,\tilde{a})$$

$$= \left(\lambda\rho_{at_1,t}(s) + (1-\lambda)\rho_{at_2,t}(s)\right)\pi(a|s)$$

$$\sum_{\tilde{a}\in\mathcal{A}}\left(\lambda\pi_{at_1}(\tilde{a}|s,a) + (1-\lambda\pi_{at_2}(\tilde{a}|s,a))\right)P_{sim}(s'|s,\tilde{a})$$

$\lambda\rho^\pi_{at_1,t}(s) + (1-\lambda)\rho^\pi_{at_1,t}(s)$ is a valid distribution, and $\mathbf{\Pi}_g$ is convex. This proves that the transition distribution at each time step is convex. The normalized discounted sum of convex sets (Equation 9) is also convex. Since the exponential discounting factor $\gamma \in [0,1)$, the sum is bounded as well. $\quad\square$

530 We now prove Lemma 4.2.

531 **Lemma 4.2.** $\overline{\mathrm{RL}} \circ \overline{\mathrm{ATIRL}}_\psi(P_{real}) = \operatorname{argmin}_{\rho_g \in \mathcal{P}_g} \psi^*(\rho_g - \rho_{real})$.

532 *Proof of Lemma 4.2.* Let $\overline{c} = \overline{\mathrm{ATIRL}}(P_{real})$, $\overline{\rho}^g = \overline{\mathrm{RL}}(\overline{c}) = \overline{\mathrm{RL}} \circ \overline{\mathrm{ATIRL}}(P_{real})$ and

$$\hat{\rho}_g = \operatorname*{argmin}_{\rho_g} \psi^*(\rho_g - \rho_{real}) = \operatorname*{argmin}_{\rho_g} \max_c -\psi(c) + \sum_{s,a,s'} (\rho_g(s,a,s') \tag{14}$$
$$- \rho_{real}(s,a,s'))c(s,a,s')$$

533 where $\psi^* : \mathcal{C}^* \longmapsto \overline{\mathbb{R}}$ is the convex conjugate of $\psi$, defined as $\psi^*(c^*) := \sup_{c \in \mathcal{C}} \langle c^*, c \rangle - \psi(c)$.
534 Applying the above definition to the rightmost term in the above equation gives us the middle term.

535 We now argue that $\overline{\rho}_g = \hat{\rho}_g$ which are the two sides of the equation we want to prove. Let us consider
536 loss function $L : \mathcal{P}_g \times \mathbb{R}^{\mathcal{S} \times \mathcal{A} \times \mathcal{S}} \longmapsto \mathbb{R}$ to be

$$L(\rho_g, c) = -\psi(c) + \sum_{s,a,s'} (\rho_g(s,a,s') - \rho_{real}(s,a,s'))c(s,a,s') \tag{15}$$

537 We can then pose the above formulations as:

$$\hat{\rho}_g \in \operatorname*{argmin}_{\rho_g \in \mathcal{P}_g} \max_c L(\rho_g, c) \tag{16}$$

$$\overline{c} \in \operatorname*{argmax}_c \min_{\rho_g \in \mathcal{P}_g} L(\rho_g, c) \tag{17}$$

$$\overline{\rho}_g \in \operatorname*{argmin}_{\rho_g \in \mathcal{P}_g} L(\rho_g, \overline{c}) \tag{18}$$

538 $\mathcal{P}_g$ is compact and convex (by Lemma B.1) and $\mathbb{R}^{\mathcal{S} \times \mathcal{A} \times \mathcal{S}}$ is convex. $L(\cdot, c)$ is convex over all $c$ and
539 $L(\rho_g, \cdot)$ is concave over all $\rho_g$. Therefore, based on minimax duality:

$$\min_{\rho_g \in \mathcal{P}_g} \max_c L(\rho_g, c) = \max_c \min_{\rho_g \in \mathcal{P}_g} L(\rho_g, c) \tag{19}$$

540 From Equations 16 and 17, $(\hat{\rho}_g, \overline{c})$ is a saddle point of $L$, implying $\hat{\rho}_g = \operatorname{argmin}_{\rho_g \in \mathcal{P}_g} L(\rho_g, \overline{c})$ and
541 so $\overline{\rho}_g = \hat{\rho}_g$.

542 $\qquad\qquad\qquad\qquad\qquad\qquad\qquad\qquad\qquad\qquad\qquad\qquad\qquad\qquad\qquad\qquad\qquad\qquad\square$

543 **B.5  Proof of Lemma 4.3**

544 **Lemma 4.3.** *The marginal transition distribution of* $\operatorname{argmin}_{\pi_g} \psi^*(\rho_g - \rho_{real})$ *is equal to*
545 $\operatorname{argmin}_{\rho_g \in \mathcal{P}_g} \psi^*(\rho_g - \rho_{real})$.

546 *Proof.* The proof of equivalence here is simply to prove that optimizing over $\pi_g$ is the same as
547 optimizing over $\rho_g$. From Equation 10 and from the fact that agent policy $\pi$ and simulator transition
548 function $P_{sim}$ are fixed, we can say that the only way to optimize $\rho_g$ is to optimize $\pi_g$, which leads
549 to the above equivalence. $\qquad\qquad\qquad\qquad\qquad\qquad\qquad\qquad\qquad\qquad\qquad\qquad\square$

# C  Experimental Details

551 To collect expert trajectories from the real world, we rollout the stochastic initial policy trained
552 in sim for 1 million timesteps, on the real world. This dataset serves as the expert dataset during
553 the imitation learning step of GARAT. At each GAN iteration, we sample a batch of data from the
554 grounded simulator and expert dataset and update the discriminator. Similarly, we rollout the action
555 transformer policy in its environment and update $\pi_g$. We perform 50 such GAN updates to ground

| Name | Value |
|---|---|
| Hidden Layers | 2 |
| Hidden layer size | 64 |
| timesteps per batch | 5000 |
| max KL constraint | 0.01 |
| $\lambda$ | 0.97 |
| $\gamma$ | 0.995 |
| learning rate | 0.0004 |
| cg damping | 0.1 |
| cg iters | 20 |
| value function step size | 0.001 |
| value function iters | 5 |

Table 1: Hyperparameters for the TRPO algorithm used to update the Agent Policy

| Name | Value |
|---|---|
| Hidden Layers | 2 |
| Hidden layer size | 64 |
| nminibatches | 2 |
| Num epochs | 1 |
| $\lambda$ | 0.95 |
| $\gamma$ | 0.99 |
| clipping ratio | 0.1 |
| time steps | 5000 |
| learning rate | 0.0003 |

Table 2: Hyperparameters for the PPO algorithm used to update the Action Transformer Policy

the simulator using GARAT. The hyperparameters for the PPO algorithm used to update the action transformer policy is provided in Table 2. The hyperparameters used for the TRPO algorithm to update the agent policy can be found in Table 1.

We implemented different IfO algorithms and noticed that there was no significant difference between these backend algorithms in sim-to-real performance. During the discriminator update step in GAIfO-reverseKL (AIRL), GAIfO and GAIfO-W (WAIL), we use two regularizers in its loss function - L2 regularization of the discriminator's weights and a gradient penalty (GP) term, with a coefficient of 10. Adding the GP term has been shown to be helpful in stabilizing GAN training [22].

In our implementation of the AIRL [10] algorithm, we do not use the special form of the discriminator, described in the paper, because our goal is to simply imitate the expert and does not require recovering the reward function as was the objective of that work. We instead use the approach Ghasemipour et al. [12] use with state-only version of AIRL.

GAT uses a smoothing parameter $\alpha$, which we set to 0.95 as suggested by Hanna and Stone [14]. RARL has a hyperparameter on the maximum action ratio allowed to the adversary, which measures how much the adversary can disrupt the agent's actions. This hyperparameter is chosen by a coarse grid-search. For each domain, we choose the best result and report the average return over five policies trained with those hyperparameters. We used the official implementation of RARL provided by the authors for the MuJoCo environments. However, since their official code does not readily support PyBullet environments, for the Ant and Minitaur domain, we use our own implementation of RARL, which we reimplemented to the best of our ability. When training a robust policy using Action space Noise Envelope (ANE), we do not know the right amount of noise to inject into the agent's actions. Hence, in our analysis, we perform a sweep across zero mean gaussian noise with multiple standard deviation values and report the highest return achieved in the target domain with the best hyperparameter, averaged across 5 different random seeds.

| Environment Name | Property Modified | Default Value | Modified Value |
|---|---|---|---|
| InvertedPendulumHeavy | Pendulum mass | 4.89 | 100.0 |
| HopperHeavy | Torso Mass | 3.53 | 6.0 |
| HopperHighFriction | Foot Friction | 2.0 | 2.2 |
| HalfCheetahHeavy | Total Mass | 14 | 20 |
| WalkerHeavy | Torso Mass | 3.534 | 10.0 |
| Ant | Gravity | -4.91 | -9.81 |
| Minitaur [38] | Torque vs. Current | linear | non-linear |

Table 3: Details of the Modified Sim-to-"Real" environments for benchmarking GARAT against other black-box Sim-to-Real algorithms.

## C.1 Modified environments

We evaluate GARAT against several algorithms in the domains shown in Figure 3. Table 3 shows the source domain along with the specific properties of the environment/agent modified. We modified the values such that a policy trained in the sim environment is unable to achieve similar returns in the modified environment. By modifying an environment, we incur the risk that the environment may become too hard for the agent to solve. We ensure this is not the case by training a policy $\pi_{real}$ directly in the "real" environment and verifying that it solves the task.

## C.2 Simulator Grounding Experimental Details

In Section 6.1, we show results which validate our hypothesis that GARAT learns an action trans-formation policy which grounds the simulator better than GAT. Here we detail our experiments for Figure 1.

In Figure 1a, we plot the average error in transitions in simulators grounded with GARAT and GAT with different amounts of "real" data, collected by deploying $\pi$ in the "real" environment. The per step transition error is calculated by resetting the simulator state to states seen in the "real" environment, taking the same action, and then measuring the error in the L2-norm with respect to "real" environment transitions. Figure 1a shows that with a single trajectory from the "real" environment, GARAT learns an action transformation that has similar average error in transitions compared to GAT with 100 trajectories of "real" environment data to learn from.

In Figure 1b, we compare GARAT and GAT more qualitatively. We deploy the agent policy $\pi$ from the same start state in the "real" environment, the simulator, GAT-grounded simulator, and GARAT-grounded simulator. Their resultant trajectories in one of the domain features (angular position of the pendulum) is plotted in Figure 1b. The trajectories in GARAT-grounded simulator keeps close to the

Figure 4: Policies trained in "real" environment, GAT-grounded simulator, and GARAT-grounded simulator deployed in the "real" environment from the same starting state

"real" environment, which neither the ungrounded simulator nor the GAT-grounded simulator manage. The trajectory in the GAT-grounded simulator can be seen close to the one in the "real" environment initially, but since it disregards the sequential nature of the problem, the compounding errors cause the episode to terminate prematurely.

An additional experiment we conducted was to compare the policies trained in the "real" environment, GAT-grounded simulator and GARAT-grounded simulator. This comparison is done by deploying them in the "real" environment from the same initial state. As we can see in Figure 4, the policies trained in the "real" environment and the GARAT-grounded simulator behave similarly, while the one trained in the GAT-grounded simulator acts differently. This comparison is another qualitative one. How well these policies perform in w.r.t. the task at hand is explored in detail in Section 6.2.