[Reviews · NeurIPS 2020]

Review 1

Summary and Contributions: This paper considers the important problem of transferring a policy learned in simulation to the real world. Specifically, the authors consider the approach of Grounded Action Transformation (GAT) for grounding a simulator in observations of the agent in the real world. Specifically, this approach learns to map state and action of the agent in the real environment to an action that will be executed in simulation but produces the same state transition as observed in the real world. The authors observe and prove that GAT is basically form of Imitation from Observations (IfO). Then the authors go ahead and use recent successes in IfO to propose a new adversarial initiation learning algorithms for action transformations. The authors show on different simulated tasks, simulators and variations of one environment that the new algorithm GARAT better reduces the domain shift the GAT. After reading the authors response, I think they addressed some of the concerns. My opinion is unchanged. A remaining concern is the 'real' naming. While I recognize the intend of the method, the authors have not shown the transfer to a real robot. Therefore, this would need to be re-worded.

Strengths: - the paper is clear and formal in deriving the algorithm and provide the proofs to backup their arguments. - they also provide the empirical evaluation to show that the new algorithm produces policies that performs better or equally well as policies learned by significantly more roll-outs. - the paper is clear and self-contained. I enjoyed reading it.

Weaknesses: Experiments on a real robot are missing. This would be the ultimate test and typically presents more challenges than any simulation environment. However, due to the current circumstance, real robot experiments may not have been possible. The authors did a good job at evaluating their approach with a broad set of experiments. A better analysis/discussion of why GAT performed so much worse from GARAT would have been desirable.

Correctness: The method and derivation seem correct.

Clarity: Very well written.

Relation to Prior Work: Seems to be well laid out. I am however not an expert in this area and may not be aware of relevant related work.

Reproducibility: Yes

Additional Feedback:


Review 2

Summary and Contributions: A new sim-to-real transfer algorithm, namely GARAT, is presented. GARAT is based on adversarial imitation from observation. The main idea of the paper is based on interacting with the target domain to make the simulator more realistic, namely grounded sim-to-real. The paper argues that previous method of ‘grounded action transformation (GAT)’[14] can be seen as an IfO method and thus IfO methods such as [43] can be effective for improving sim-to-real transfer. Based on that, it is proposed that IfO approaches can be repurposed for sim-to-real. Experimental evaluation is done on several simulation domains with different dynamics and it is reported that GARAT performs better than black-box sim-to-real methods.

Strengths: + mathematical proof is provided to show that it is possible to learn action transformation policy directly from data. + Sim-to-Real topic which is an important topic in policy learning is considered in this paper. + Posing Sim-to-Real as an IfO is interesting.

Weaknesses: -The cost function formulation is similar to previous work of [17] and [43]. The adversarial objective minimized is based on prior work of [17] and [43]. Given this, the proposed approach does not offer significant technical novelty. - Throughout the paper, it is said that the proposed method a sim-to-real approach and the transferred domain is called ‘real’. However, the experiments are based on sim-to-sim evaluation where there are two simulator for a task and one of them is called ‘real’. I do not see such characterization as acceptable. If the proposed approach is a transfer learning approach for transferring between two domains it should not be called sim-to-real. Sim-to-real research seeks to present techniques that are trained in simulation but will eventually be tested and evaluated on *real physical* robots/domains (please see references [r.1, r.2, r.3, r.4, r.5, r.6] provided below) It is more correct to call this work as a sim-to-sim or transfer learning approach. - The experimental results are conducted on limited domains only including several MuJoCo environments and Mintaur. What would be the performance of the propose approach if high dimensional observation (such as images) is considered? -In the experiment of Fig. 1(b), why is the curve of GAT cut at around 25 in Time? What would be the result of GAT otherwise? -In the experiment of Fig. 3, why is the performance of GARAT is sometimes more that upper bound scaled reward of 1 (the rewards are scaled such that \pi_{real} achieves 1) ? ~~~~Post Rebuttal Comments~~~~~ My main concern about this paper is that the technical novelty is not significant and the experimental evaluations and comparisons, whether sim-to-real or sim-to-sim, are not sufficient to demonstrate the effectiveness and the weaknesses of the proposed approach or backup the claim the the proposed approach can “be successfully adapted to sim-to-real problem.” . I also share the concerns of the other reviewers that more exhaustive experiments are required and even for sim2sim experiments different simulation environments should be considered. Unfortunately, these concern is not well addressed in the rebuttal. Also, my other question about high dimensional observation is not addressed. Conditioned on commitment from the authors that they modify the paper from “sim-to-real” to “sim-to-sim” within the current manuscript and the commitment and to add the missed references, I update my rating.

Correctness: I did not notice any explicit error. For comments and questions about method and experiments please see other sections.

Clarity: -The presentation of the paper can be improved. I found the paper rather hard to follow although most of the sections were re-explanation of past works (such as sec. 2.1, sec. 2.2, sec 2.3, and parts of sec. 4). I’d suggest, re-organizing the paper such that the main contributions be better highlighted and re-explanation of past work be moved to appendix to help the reader better distinguish what is the paper contribution and novelty and what is the recap of the past known methods. - Fig. 2 is not much informative, I’d suggest that space be used to add some of the other experimental results from the appendix.

Relation to Prior Work: - While the main topic of the paper is explained to be sim-to-real, some successful past work on sim-to-real are not mentioned. Experimental evaluations also misses comparison with some of successful past works, such as domain randomization[32,39]. To name a few of miss citations, please check the references [r.1, r.2, r.3, r.4, r.5, r.6] listed below. Adding citation to [r.1, r.2, r.3, r.4, r.5, r.6] is strongly recommended in any version of the manuscript. [r.1]. Stephen James, Andrew J Davison, and Edward Johns. Transferring end-to-end visuomotor control from simulation to real world for a multi-stage task. Conference on Robot Learning, 2017. [r.2]. Fereshteh Sadeghi, Alexander Toshev, Eric Jang, and Sergey Levine. Sim2real viewpoint invariant visual servoing by recurrent control. In IEEE Conference on Computer Vision and Pattern Recognition, 2018. [r.3]. Jan Matas, Stephen James, and Andrew J Davison. Sim-to-Real reinforcement learning for deformable object manipulation. Conference on Robot Learning, 2018. [r.4]. Konstantinos Bousmalis, Alex Irpan, Paul Wohlhart, Yunfei Bai, Matthew Kelcey, Mrinal Kalakrishnan, Laura Downs, Julian Ibarz, Peter Pastor, Kurt Konolige, Sergey Levine, and Vincent Vanhoucke. Using Simulation and Domain Adaptation to Improve Efficiency of Deep Robotic Grasping. IEEE Intl. Conference on Robotics and Automation, 2018. [r.5]. Stephen James , Paul Wohlhart , Mrinal Kalakrishnan , Dmitry Kalashnikov , Alex Irpan , Julian Ibarz , Sergey Levine , Raia Hadsell , Konstantinos Bousmalis, Sim-to-Real via Sim-to-Sim: Data-efficient Robotic Grasping via Randomized-to-Canonical Adaptation Networks, CVPR, 2019. [r.6]. Stephen James, Michael Bloesch, and Andrew J Davison. Task-embedded control networks for few-shot imitation learning. Conference on Robot Learning, 2018.

Reproducibility: Yes

Additional Feedback:


Review 3

Summary and Contributions: Post rebuttal I would like to thank the authors for their comprehensive response. I agree that the proposed (sim2sim) method is valuable and can be eventually used for sim2real. However, I think the current paper should be about "sim2sim transfer" or "adaptation to changes in dynamics" without demonstrating sim2real results; therefore, the title of "Towards Sim-to-Real Transfer: ..." is still an overclaim. Moreover, extensive experiments on environments with dynamics changes in many properties are needed to strengthen the claims. I will stick to a weak accept here. ----------------------------------------------------------------------- The paper proposes to formulate the problem of grounded action transformation, which learns to transform a policy to match the dynamics of the target environment for sim-to-real transfer, as imitation learning from observation (IfO). This grounded action transformation can be done by constructing a new MDP with a new state space with the joint state and action space of the original MDP. Then, an expert trajectory in the original MDP corresponds to a sequence of states in the target MDP, which can be used for IfO. With IfO formulation, the proposed method learns the action transformation using IfO techniques. The empirical results demonstrate that the grounded action transformation using IfO adapts a policy to the changes in dynamics only with a few thousands of real-world interactions.

Strengths: - The paper poses the grounded action transform as imitation from observation, which is novel and opens up the possibility of using IfO algorithms for transfer learning. - The experimental results verify efficient adaptation of a policy from one environment to another environment with different dynamics. - The theoretical derivation of IfO formulation is clear and correct. - The implementation of the proposed method is simple and straightforward.

Weaknesses: - The empirical evaluation is not enough to claim the applicability of the proposed method to general sim-to-real transfer. Since the proposed method is built upon grounded action transformation, it does not consider discrepancy in the state space and noise, which is a critical issue in sim-to-real transfer. The real-world experiments or more exhaustive experiments in simulation are required. - Experiments have conducted mostly on locomotion benchmarks. To support the claim of sim-to-real transfer under a general setup, it would be necessary to demonstrate results on robotic manipulation tasks, such as openai gym robotics environments and robosuite environment. - As can be seen in the Figure 1(b), the error between GARAT and "real" exists in a simple environment. How can RL reduce the gap present in the action transformation?

Correctness: The proposed method and experiments are correct.

Clarity: The paper is generally clear and well written.

Relation to Prior Work: The paper clearly discusses difference between the proposed method and prior work.

Reproducibility: Yes

Additional Feedback: - The paper includes most details to reproduce the results. - Figure 1(b): How many "real" trajectories and policy updates are used to get the plot? - Which variant of IfO algorithms is used for the main results? - What is the rationale behind the choice of TRPO and PPO for training agent policy and action transformation policy, respectively? - Some curves in Figure 3 show decreasing performance as trained with more samples which is counter-intuitive. - As an ablation, fine-tuning the agent policy (without action transformation) can demonstrate the efficacy of grounded action transformation. Minor comments: - L104: can also been seen as -> can also be seen as - Figure 1(b): The GAT curve stops in the middle.


Review 4

Summary and Contributions: The paper proposes a sim2"real" transfer framework. The paper extends Grounded Action Transformation into Generative Adversarial into Generative Adversarial Reinforced Action Transformation by using adversarial imitation techniques. They then demonstrate the GARAT is better then GAT at transforming.

Strengths: They do show that they can transfer policies between two simulators (OpenAI Gym and Mujoco) and show that they can transfer between varying transition dynamics within the same task. This paper shows that it can adapt to different parameters in the simulation environment effectively. For instance, parameters such as gravity, mass, and friction can be varied and GARAT can adapt to these new parameters much quicker than GAT. This paper would benefit the community for those that currently use GAT. This paper demonstrates contributions to the problem of sim-to-sim performance transfer and ways to make agents generalize to various parameters in simulation. The method in the paper is novel and relevant for the community for sim-to-sim transfer.

Weaknesses: The biggest weakness of this paper is that it claims to do sim-to-real tasks, but in reality the tasks are sim-to-sim. The paper does not demonstrate that transferring between two physics simulators which approximate physics in a similar albeit different way, is not a kin to transferring to a real world policy. GARAT's performance remains untested on the actual problem of sim-to-real transfer. This paper demonstrates that GARAT may perform better than GAT in sim-to-sim. Even the word "real" is quoted in experiments listed in the end of the paper, but not in the title or abstract or introduction. The paper has demonstrated sim-sim Even for sim2sim experiments, I would like to see more than one between entirely different simulators. This is a different type of generalization that simply adjusting to different parameters within the same simulation. These parameters that need to be adjusted to may even vary over time due to declining battery levels in the real world. Due to these differences between sim-to-sim transfer and deploying these simulation learned policies on actual robots, this paper does not actually demonstrate sim-to-reality transfer. Rather, it only demonstrates sim-to-sim transfer.

Correctness: They are reasonable correct for sim2sim experiments, however, they are missing any "real" experiments for reality. The methods therefore do not back the claims of sim-to-real transfer, but rather only sim-to-"real" transfer.

Clarity: Reasonably so.

Relation to Prior Work: Yes

Reproducibility: Yes

Additional Feedback: This paper should either be reframed as a sim-to-sim paper or have at least one experiment in reality to demonstrate that it can actually transfer from sim-to-reality. The current global pandemic may make real world robotics experiments difficult, but that should affect both the claims and the methods of the paper, not just the methods. Update: Given the author's rebuttal and promised changes in the final version, I have decided to increase my score accordingly.

[Author Response · NeurIPS 2020]

We thank the reviewers for lending their expertise and time to provide feedback on our efforts. We are glad that all the
reviewers found our insight that action transformation can be seen as an IfO problem novel and interesting. We respond
to the biggest questions and comments below and will address all feedback in the paper.

[R2 , R3 , R4 ] The reviewers are correct in pointing out that, despite the title, we do not include a real robot experiment.
Our work is motivated by sim-to-real, but we were unable to conduct real robot experiments due to the current pandemic
as R1 and R4 pointed out. If accepted, we will make several changes to moderate the claims as R4 suggested. In
particular, we will change the terminology in the paper to align with more general transfer learning, using source
and target domains as opposed to sim and "real." Also, we will change the title to "Towards Sim-to-Real Transfer:
An Imitation from Observation Approach." Please note that our formulation remains very relevant to the sim-to-real
community. We would like to highlight that one of our experiments is indeed an excellent proxy for the sim-to-real
problem: In the Minitaur domain (Figure 2), Tan et al. [38] found that while their existing simulator (our source domain)
inaccurately represented their robot, the new simulator they crafted (our target domain) *did* enable direct policy transfer
from sim to real.

[R2 , R3 ] Both manipulation domains [7, 24, 26, 39, 40, R2 's suggestions] and locomotion domains [9, 10, 11, 14,
18, 27, 38, 46] are prevalent in the sim-to-real literature. Both are important—but different—problems: manipulation
domains are more likely to exhibit observation mismatch, whereas locomotion domains are more typically associated
with dynamics mismatch. The scope of our work here is mainly dynamics mismatch, and therefore we focus our
experiments on locomotion problems. GARAT solely addresses dynamics mismatch. For locomotion, the observations
are usually joint angles and velocities, so observation mismatch is negligible. If accepted, we will make this scope more
clear in the camera-ready version of our paper and include the references R2 suggested. Note that in our problem setting,
the state spaces are the same in the source and target domains, as is commonly the case in sim-to-real. Specifically, we
consider dealing with embodiment mismatch to be beyond the scope of this paper.

[R2 ] Most domain randomization techniques, and all the papers suggested by R2 , require a modifiable simulator and
substantial domain expertise [7]. In this paper we focus on the case where the simulator cannot be modified (black box),
and hence it is not appropriate to compare with methods that can adjust the simulator itself. We compare to ANE [20]
which is an action randomization technique.

[R2 ] Respectfully, we strongly disagree with the reviewer's assertion that our approach does not offer significant
technical novelty. In this work, we show how tools developed in the imitation learning community can be successfully
adapted to sim-to-real problems. Moreover, our adaptation of one such tool actually leads to better performance
than alternative applicable approaches. To the best of our knowledge, this is the first time this has been studied
in the literature, and therefore our work represents a novel and important connection between two largely separate
communities.

[R1 ] Concerning why GAT was not as effective as GARAT on transfer, perhaps it would be useful to compare the two
techniques to their imitation learning equivalents, behavioral cloning (BC) and using inverse RL (IRL). BC suffers
from distribution shift while IRL methods are able to learn how to recover from such shifts; likewise, GAT is unable to
recover from the shift introduced by an imperfect action transformation while GARAT can correct for such deviations.
GAT is myopic, trying to match single transitions, while GARAT matches the whole trajectory (Figure 1b).

[R2 , R3 ] The curve for GAT cuts off early in Figure 1b. In the InvertedPendulum domain, the episode terminates if
the angle of the pendulum exceeds $\pm 0.2$ radians. In the environment with GAT, the action transformation learns to
keep close to the target domain's dynamics early on, but this causes instability later in the episode, leading to early
termination. GARAT sacrifices initial accuracy to keep the overall trajectory as realistic as possible. We will edit the
caption for Figure 1b to make it clear in the camera ready version of the paper.

[R3 ] We use the loss derived in Section 4.3 in our main results. Our algorithm is agnostic to the RL algorithms used
for training. We chose PPO and TRPO for the action transformation function and the agent respectively because that
combination worked best in preliminary experiments on the InvertedPendulum domain.

[R3 ] GARAT should implicitly address process noise due to its adversarial learning procedure. The discriminator in
GARAT encourages the action transformation function to learn a distribution of transitions that are similar to the target
domain, including any noisy transitions. Moreover, GAT [14] has been shown to be useful in sim-to-real transfer on a
real legged humanoid robot, showing that impact dynamics and operational noise do not prevent learning.

[R2 , R3 ] Figure 3 was normalized in order to compare the performance of different algorithms across different
domains. It does not represent the maximum and minimum returns possible. We train $\pi_{real}$ in the target domain for 1
million time-steps, enough to reach a reasonable policy. These policies may take more training to converge completely
(HalfCheetah is usually trained for 10 million timesteps). GARAT manages to learn a policy that does better than the
policies trained directly in the target domain for some of these environments.

[Meta-Review · NeurIPS 2020]

Summary: This paper proposes a new technique for learning to transfer optimal policies obtained from a simulator to a real world environment. The only difference between sim and real is in the state transition probabilities. The main idea consists in learning an action grounding function that maps state-actions learned in simulation to modified actions that are executed in the real system. The authors notice that this problem is similar to a variant of imitation learning, where the imitator learns to match state trajectories (where the actions are unknown) demonstrated by an expert. Experiments on MuJoCO where the "real" environment is obtained by modifying physical properties (such as mass and friction) from their values in simulation. Pros: - Nice connection between transfer learning and inverse reinforcement learning - Theoretical guarantees about the optimality of the transfer Cons: - The use of the term real is misleading. There are no real experiments here, which makes the evaluation weak, especially since this is supposed to be about sim2real. Discussion and decision: The discussion is centered around the use of the term real in the paper, which is highly misleading. Any work on sim2real must be evaluated on real environments. There are concerns from a reviewer about the similarities between the objective function in this paper and previous ones such as [43], but this work is concerned with transfer learning and not imitation learning as in previous works. The re-use of previous tools can be seen as an original application to other domains. NeurIPS does not have conditional accept mechanisms. But if this paper is accepted, the authors must remove the term "real" from the paper. The area chairs and all the reviewers find the use of this term inaccurate and highly misleading to researchers in robotics. I suggest to use terms such as transfer learning, or domain adaptation or even sim2sim. Potential sim2real experiments can be discussed and used as a motivation behind this work, but this paper cannot claim that it proposes a sim-2-real method without backing up the claim with actual sim-2-real experiments.